# Eu^3+^-Doped (Gd, La)AlO_3_ Perovskite Single Crystals: Growth and Red-Emitting Luminescence

**DOI:** 10.3390/ma16020488

**Published:** 2023-01-04

**Authors:** Tong Wu, Qian Zhang, Yun Shi, Ling Wang, Yifei Xiong, Hui Wang, Jinghong Fang, Jinqi Ni, Huan He, Zhenzhen Zhou, Qian Liu, Jianding Yu

**Affiliations:** 1State Key Laboratory of High Performance Ceramics and Superfine Microstructure, Shanghai Institute of Ceramics, Chinese Academy of Sciences, Shanghai 200050, China; 2College of Physical Science and Technology, Central China Normal University, Wuhan 430079, China; 3Center of Materials Science and Optoelectronics Engineering, University of Chinese Academy of Sciences, Beijing 100049, China

**Keywords:** Eu:GdAlO_3_ crystal, optical floating zone method, red emission, plant lighting, X-ray imaging

## Abstract

Eu^3+^-doped GdAlO_3_ (Eu:GAP) and Gd_0.5_La_0.5_AlO_3_ (Eu:GLAP) perovskite single crystals were successfully grown using the optical floating zone (OFZ) method. The microstructure, optical, photoluminescence (PL) and radioluminescence (under X-ray excitation, XEL) were investigated. Under the PL excitation of 275 nm, obvious emission bands peaking at 556 nm, 592 nm, 617 nm, 625 nm, 655 nm, and 706 nm were demonstrated, which correspond to the 5D_0_ → 7F_j_ (j = 0–4) transitions of Eu^3+^. The grown Eu:GAP single crystal showed a stronger PL intensity compared with that of Eu:GLAP in the red light region. After annealing at 1000 °C for 4 h in weak reductive atmosphere (Ar + 5% H_2_), a slight redshift and dramatic enhancement of PL and XEL intensity occurred. In addition, Eu:GLAP show a more intense XEL emission than that of Eu:GAP. Considering their different densities, these two kinds of red luminescence phosphors are proposed to be promising in a wide field of X-ray imaging, warm white, or plant lighting, respectively.

## 1. Introduction

Rare earth perovskite oxides (REMO_3_; RE = Y, La, Lu; M is metal) have demonstrated their advantages in the fields of electricity, magnetism, and photocatalysis, etc. [1,2,3], while rare earth aluminate perovskites (REAlO_3_) are relatively promising as a luminescence host [4]. In REAlO_3_, the lattice position of RE is surrounded by twelve O^2−^ anions and can be replaced by various RE3+ ions, where RE = Gd, Y, or Lu [5,6]. Among these, the REAlO_3_ perovskite compound formed by Gd^3+^ with a large ion radius (180.4 pm) [7] is close to the cubic crystal cell (ideal perovskite structure, Pm3m), and the symmetry is higher than that of YAlO_3_ (YAP) and LuAlO_3_ (LuAP). Thus, the high accommodation capacity of the perovskite structure helps to regulate electronic and spectroscopic properties through the substitution of Gd3+ in GAP by various activators, such as Eu^3+^, Er^3+^, Ce^3+^, Yb^3+^, and Dy^3+^ [8,9]. Meanwhile, the short optical cut-off length, high absorption cross section, and large atomic weight are also conductive to achieve strong and fast emissions under excitations [10].

Among the rare earth ions, Eu^3+^ is characterized by high luminescence intensity and milliseconds of lifetime, and considered a promising activator in red phosphors or scintillators [11,12,13,14]. In addition, in the distorted GdO_12_ polyhedral, the sites occupied by Eu^3+^ ions lack inverse symmetry and satisfy both magnetic and electric dipole transitions [15,16]. In 2018, Gorbenko et al. [2] reported the epitaxial growth of Eu^3+^-doped RAlO_3_ (R = Y, Lu, Gd, Tb) single crystalline film for scintillating screens, finding that the Gd^3+^ → Eu^3+^ energy transfer improved the luminescence efficiency of Eu^3+^. In addition, the high segregation coefficient of Eu^3+^ ions in GAP, i.e., 0.97–1.03, indicated that Eu^3+^ can be accommodated in a wide concentration range. Furthermore, GAP has a low thermal expansion coefficient [17] and high thermal conductivity [18,19,20], which is suitable for crystal growth.

Based on composition engineering, cation substitution was used to modify the crystal field and the energy level positions of the activators and traps in the bandgap. A considerable enhancement of red emissions was achieved by co-doping Tb^3+^ [9,21], Dy^3+^ [8,22], or Yb^3+^ [23] to Eu:GAP phosphors. However, composition engineering in GAP, such as Eu:(M,Gd)AlO_3_ (Eu:MAP, M = La, Lu, Y), was rarely reported. The composition regulation has shown a significant effect on the scintillation properties in the garnet structures Gd_3_Al_5_O_12_ (GAG) [24,25,26]. The modification of host components is mostly to replace Gd sites with lanthanide ions, such as LaAlO_3_ (LAP) with space group Pm3m (No. 221), which shows an extremely stable structure at room temperature [27]. However, it cannot be considered as a traditional host material because it lacks a 4f electron layer [28].

Eu:GAP powder phosphors have been widely studied. Broadband or line-spectral luminescence in multiple bands in the 590–696 nm range can be observed in Eu:GAP phosphor synthesized by various methods, such as the sol–gel process, co-precipitation method, combustion synthesis, and precursor decomposition approach [29,30,31]. Luo et al. [32] investigated the Eu concentration effect in GAP powder phosphors and concluded that the 8 at.% Eu-doped GAP reached the maximum luminescence intensity. Compared with powders, the single crystal has high lattice integrity because of the low defect concentration inside it [33], helping to achieve high luminescence efficiency. However, the growth of GAP crystals has rarely been reported so far.

We investigated the growth of 8 at.% Eu-doped GAP and GLAP single crystals using the optical floating zone (OFZ) method in this work. The optical quality and photoluminescence (PL) properties of the single crystals were investigated in detail. Furthermore, the radioluminescence was also characterized by X-ray excitation.

## 2. Materials and Methods

### 2.1. Crystal Growth

The starting high-purity raw materials (≥99.99%, Eu_2_O_3_, La_2_O_3_, Gd_2_O_3_, α-Al_2_O_3_) were weighed according to the chemical stoichiometric of (Eu0.08LaxGd0.92-x)AlO_3_, where x = 0, 50%, i.e., Eu_0.08_Gd_0.92_AlO_3_ and Eu_0.08_Gd_0.5_La_0.42_AlO_3_, respectively. Then, they were mixed using an automatic agate mortar and an additional 1 wt.% B_2_O_3_ was used as the flux. The mixed powders were calcined and molded to rods with a dimension of Ø 8 mm × 100 mm for cold isostatic pressing treatment under 70 MPa for 20 min. The as-prepared rods were sintered to the feed ceramics rod in the tube furnace at 1600 °C for 8 h with flowing O_2_.

The optical floating zone furnace (FZ-T1000H CSC Cop., Tokyo, Japan) was used for the crystal growth. The feed rod was fixed with platinum wire on the upper side, and the seed crystal was fixed with nickel wire on the lower side. Firstly, we used a GAP ceramic rod as a seed rod, and then the obtained GAP crystal was used as the seed for the successive growth. During the growth process, the feed and seed rods rotate in opposite directions to stir the molten zone sufficiently. The growth rate (i.e., the moving speed of the halogen lamp stand) was set to 5 mm/h and the rotation speed was set to 20 rpm. High-purity (100%) oxygen was used as the growth atmosphere to help to reduce the concentration of defects such as oxygen vacancy. The heat source was aligned with the lower end of the feeding rod to realize inoculation with the seed rod after melting, and then the solid–liquid interface moved continuously with the platform (heat source) to realize the gradual solidification of the melt and finally complete the equal diameter growth. Partial crystals cut into pieces were put into the tube furnace for annealing in flowing Ar (+5% H_2_) at 1000 °C for 4 h. Double-sided crystal blocks polished to size Ø 6 mm × 3 mm were used for the absorption spectrum measurement.

### 2.2. Characterization

To determine the phase purity and crystal structure, the as-grown crystals were ground into uniformly distributed powders for powder X-ray diffraction (PXRD) measurement at room temperature using the Rigaku Ultima IV diffractometer (Cu Kα, 40 kV, 40 mA, Rigaku Ultima IV, Tokyo, Japan). The scanning speed was 5°/min in the range of 20~90°. The absorbance spectra of the crystals were measured using a UV−Vis−NIR photometer (Varian Cary 5000, Cary 5000, Agilent Technologies Inc., Santa Clara, CA, USA), and the apparatus baseline was subtracted automatically. PL and excitation (PLE) spectra were recorded using a fluorescence spectrometer (Hitachi F-4600, Tokyo, Japan). The steady-state radioluminescence XEL spectra of the samples were measured using the self-assembled X-ray fluorescence spectrometer (X-ray tube: 70 kV, 1.5 mA, Ruhai Cop., Shanghai, China).

## 3. Results

Figure 1a shows a photograph of the as-grown Eu:GAP and Eu:GLAP crystals rods. All of the crystals are almost crack-free and are of a dark brown color. Figure 1b shows the crystal circle pieces cut vertically along the direction of crystal growth. After annealing in flowing Ar (+5% H_2_), the Eu:GLAP crystal pieces appeared transparent, while the Eu:GAP crystals were an opaque white.

Figure 2 shows the PXRD patterns of the as-grown Eu:GAP and Eu:GLAP crystals. All diffraction peaks of the crystals matched well with the standard GAP (PDF#46-0395), indicating a formation of the desired perovskite GAP phase. All crystals show a similar crystal structure and belong to the cubic crystal pm3m space group. In addition, the absence of secondary phases indicates that Eu^3+^ completely entered into the GAP lattice through a solid solution [21]. In addition, the diffraction peak did not present a dramatic shift, which might indicate that the lattice constants of Eu:GAP and Eu:GLAP crystals are similar.

Figure 3 gives the Laue photograph of the Eu:GAP and Eu:GLAP crystals, as grown and after the air annealing process. The clear and well symmetric diffraction patterns demonstrated a high single crystal quality.

In Figure 4, the absorbance spectra of the Eu:GAP and Eu:GLAP crystals with and without the Ar (+5% H_2_) atmosphere annealing process are presented. The strong absorption in the 200–350 nm range is ascribed to the characteristic absorption of the Eu-O charge transfer band (CTB), corresponding to the electron transition from 2p of O^2−^ to Eu^3+^ 4f, indicating that Eu^3+^ has successfully partially replaced the octa-coordinated Gd^3+^ ion site. The weak absorption peaks appearing at 320 nm are related to 8S_7/2_ →6P_j_ of Gd^3+^. Its 6P_j_ energy level can be used as a platform for transmitting energy to the higher energy level 4f6 of Eu^3+^ ion to improve the energy transfer efficiency. In addition, the sharp and weak absorption peak at approximately 400 nm is related to the 4f-4f transition of Eu^3+^. It is also worth noting that the redshift of the absorption cutoff occurred in Eu:GAP crystals regarding Eu:GLAP crystals, which reveals the band gap difference between them. The absorbance of the Eu:GAP crystals in the 400–800 nm range is still around 2, which indicates a low transparency of the crystals. We suspect that the small number of inclusions and cracks might be the possible light scattering sources that introduced a decrease in the optical transmittance. The crystal growth parameters should be investigated carefully in the future to improve the optical quality, axial homogeneity, and radial homogeneity of the as-grown crystals.

The PLE (λ_em_ = 617 nm) and PL (λ_ex_ = 275 nm) spectra of the Eu:GAP and Eu:GLAP crystal powders are shown in Figure 5, respectively. All samples exhibited a similar spectral line and can be considered to possess the same luminescence phenomenon. As shown in Figure 5a, the excitation peaks of the Eu:GAP and Eu:GLAP crystals matched well with each other in terms of the wavelength position, although there was a difference in the intensity. The emission in the 200–350 nm range originated from the O^2−^-Eu^3+^ charge transfer band (CTB) transition. In addition, the excitation peaks at 360 nm (7F_0_ → 5L_0_), 373–383 nm (7F_0_ → 5G_3_), 396 nm (7F_0_ → 5L_6_), 415 nm (7F_0_ → 5D_3_), and 466 nm (7F_0_ → 5D_2_) can be attributed to the 4f-4f transition of Eu^3+^ ions [20]. Figure 5b shows the emission spectra recorded under 275 nm excitation. The emission of the Eu:GAP single crystal in the red region is stronger than that of the La co-doped crystal and the emission spectra of all samples are the 4f characteristic sharp line emission of Eu^3+^, i.e., 5D_0_ → 7F_J_ (J = 0, 1, 2, 3, 4) transition. Among them, the emission at 592 nm belongs to the 5D_0_–7F_1_ transition allowed by the magnetic dipole without an electric dipole contribution. The red emission at 617 nm and 625 nm only originates from the 5D_0_ → 7F_2_ electric dipole transition of Eu^3+^. Unlike magnetic dipole transitions, electric dipole transitions with higher transition strength are highly sensitive to the local environmental symmetry around Eu^3+^ ions [34]. The emission at 556, 655, and 706 nm results from 5D_0_ → 7F_0_, 5D_0_ → 7F_3_ and 5D_0_ → 7F_4_ transitions [35,36]. A near-infrared (NIR) emission at 836 nm was also observed.

In Figure 5b, the significantly improved PL intensity of the as grown Eu:GAP crystal regarding other crystals can be attributed to the enhanced crystallinity or possibly higher effective Eu concentration, since the crystals will exhibit axial or radial inhomogeneity in different positions of the crystal rods [37,38]. In this work, we only cut the part that had good crystal integrity and used it for the following measurement. This means that the selected crystals came from random positions. In addition, considering that the wavelength range that human eyes can perceive is located at 400–700 nm [39], the 556 and 655 nm emission of Eu:GAP and Eu:GLAP indicated a potential application in high-efficiency warm white lighting. Significantly, their broad emission at 655 nm and 706 nm matched well with the absorption band of red/phytochrome (PR) and far-red/phytochrome (PFR) of plants [40], indicating the potential plant lighting application of Eu:GAP and Eu:GLAP.

Figure 6 recorded the temperature evolution of PL spectra under an excitation of 275 nm. The investigated temperature range was from room temperature (RT) to 225 °C. The PL intensity varied with the increase in the temperature, which can be attributed to the thermal quenching effect. Figure 7 reveals the temperature dependence of PL integral intensity in a clearer way. The Eu:GLAP crystal exhibited a better thermal stability. Therefore, considering the homogeneity effect in different positions of the crystal rods, as we discussed in Figure 1 and Figure 5, a further systematic study on La substitution concentration to Gd, i.e., La and Gd ration in Eu:LGAP, needs to be conducted.

Figure 8 shows the Commission Internationale de L’Eclairage (CIE) chromaticity diagram of the Eu:GAP and Eu:GLAP crystals. The color coordinates are shown in Table 1. The color coordinates of the Eu:GAP crystal are (0.4903, 0.507), located in the orange-yellow region. After annealing, the color coordinates of the Eu^3+^ ions move towards the orange-red region for crystals with Eu^3+^ single-doping and Eu^3+^/La^3+^ co-doping, which gives it important application potential in warm white lighting. In addition, the application potential of the Eu:GAP and Eu:GLAP crystals in radiation detection are also explored.

The XEL spectra of the Eu:GAP and Eu:GLAP crystals with or without the 1000 °C for 4 h in Ar + 5% H_2_ annealing process are shown in Figure 9, with a Bi_4_Ge_3_O_12_ (BGO) single crystal as a reference sample. The typical 5D_0_ → 7F_J_ transition of Eu^3+^ under X-ray excitation was almost the same as that of the PL (λex = 275 nm) spectra. The significantly increased strength of the annealed crystals is expected to make them a potential candidate for applications in X-ray detection and imaging.

## 4. Conclusions

Eu:GAP and Eu:GLAP single crystals were successfully grown using the OFZ method, which had rarely been reported before. The XRD patterns show that the as-grown crystals are the GAP phase with the integral cubic perovskite structure. All crystals show a strong absorption in the 200–300 nm range, originating from the characteristic absorption of the Eu-O charge transfer band (CTB). The obvious absorption cutoff occurs at 300 nm in GLAP. In contrast, the Eu:GAP single crystal has a stronger emission in the red region and an obvious strong emission band under the 275 nm excitation located at 556 nm, 592 nm, 617 nm, 625 nm, 655 nm, and 706 nm corresponding to the 5D_0_ → 7F_0_, 5D_0_ → 7F_1_, 5D_0_ → 7F_2_, 5D_0_ → 7F_3_, and 5D_0_ → 7F_4_ transitions of Eu^3+^. The CIE chromaticity coordinates shows the emitted color shifts from the orange-yellow region to the orange-red region for the annealed crystal. Except for the application value in the field of solid-state lighting, the intense luminescence phenomenon under X-ray excitation demonstrated its potential application in the radiation detection field.

## Figures and Tables

**Figure 1 materials-16-00488-f001:**
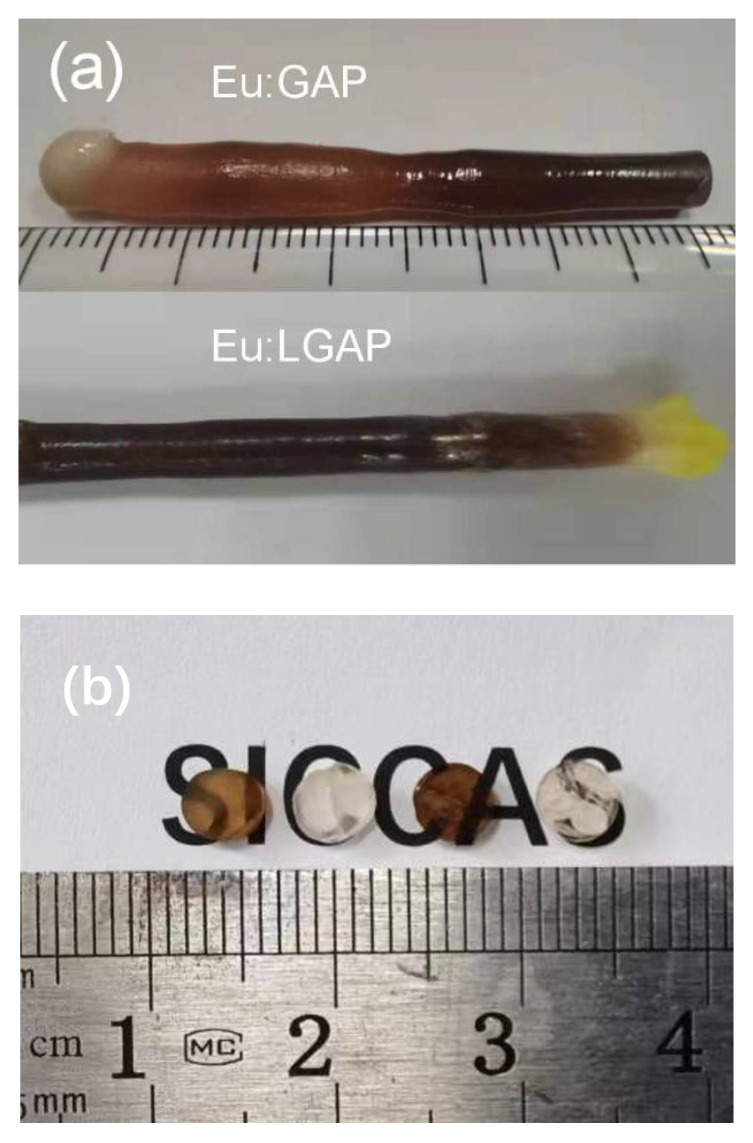
Photographs of the (**a**) as-grown Eu:GAP and Eu:GLAP crystal rods; (**b**) double face polished Eu:GAP and Eu:GLAP single crystals, from left to right, as-grown Eu:GAP and annealed Eu:GAP crystals, as-grown Eu:GLAP and annealed Eu:GLAP crystals, 1 mm thickness.

**Figure 2 materials-16-00488-f002:**
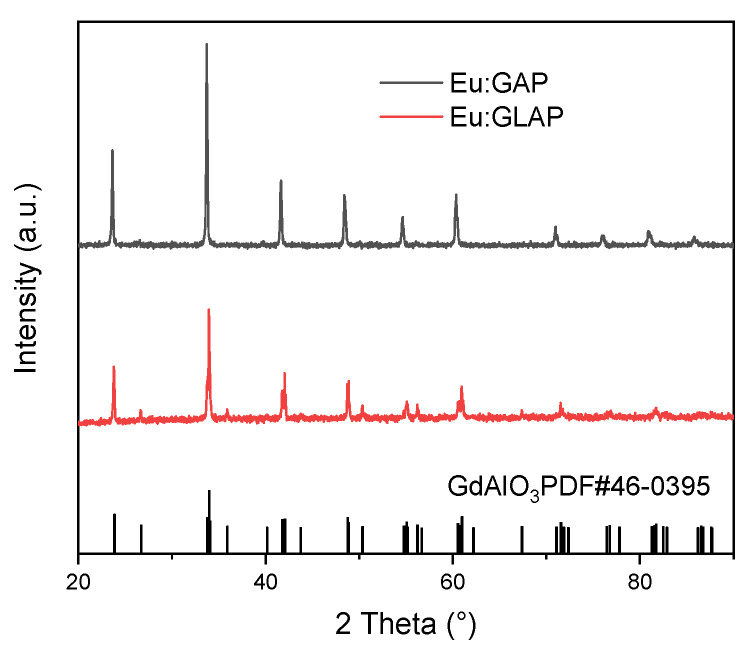
PXRD patterns of the as-grown Eu:GAP and Eu:GLAP single crystals.

**Figure 3 materials-16-00488-f003:**
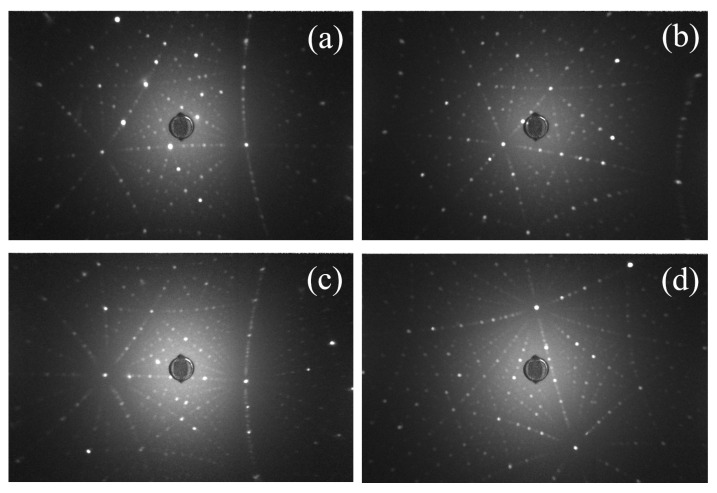
Laure graph of (**a**) as-grown Eu:GAP crystal; (**b**) annealed Eu:GAP crystal; (**c**) Eu:GLAP single crystals; (**d**) annealed Eu:GLAP crystal.

**Figure 4 materials-16-00488-f004:**
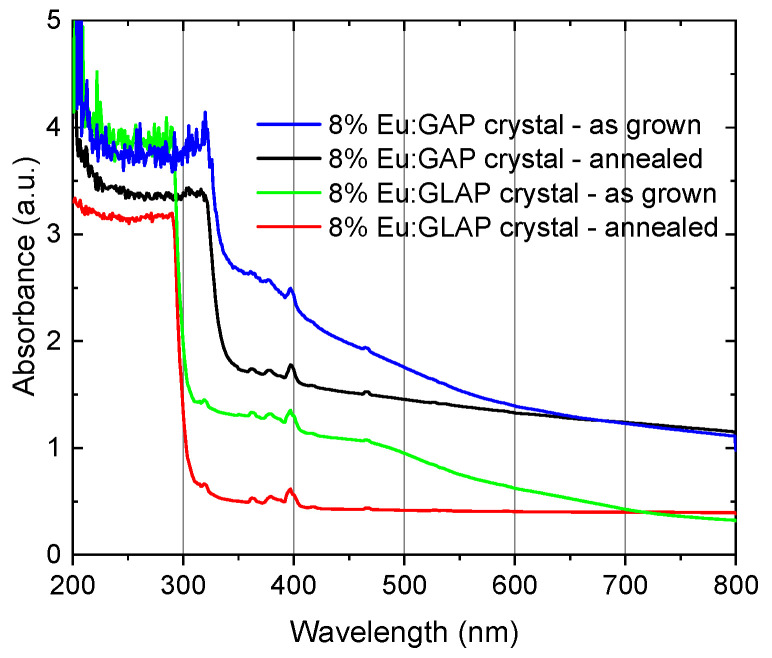
Absorbance spectra of the as-grown and annealed Eu:GAP and Eu:GLAP crystals (after double-side polishing to 1.0 mm thickness).

**Figure 5 materials-16-00488-f005:**
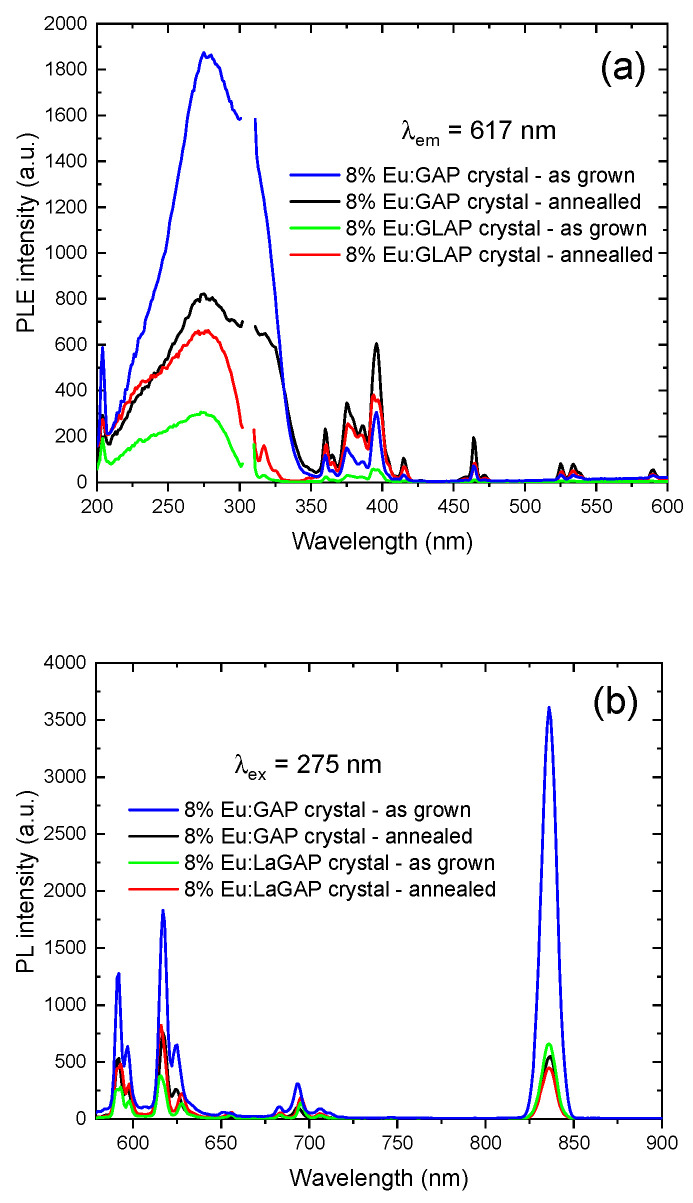
Photoluminescence spectra of the as-grown Eu:GAP and Eu:GLAP crystals; (**a**) PLE λ_em_ = 617 nm; (**b**) PL λ_ex_ = 275 nm.

**Figure 6 materials-16-00488-f006:**
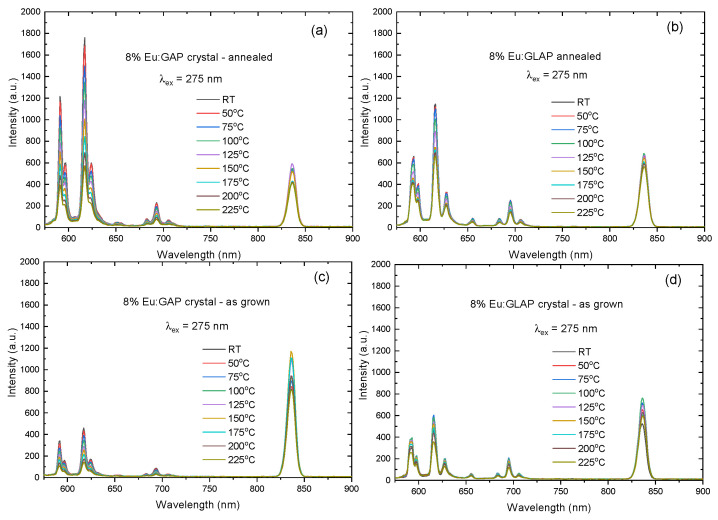
PL spectra of the (**a**) Eu:GAP crystal; (**b**) Eu:GLAP crystal with the post Ar (+5% H_2_) annealing process; (**c**) as-grown Eu:GAP crystal and (**d**) as-grown Eu:GLAP crystal at different temperatures from RT to 225 °C.

**Figure 7 materials-16-00488-f007:**
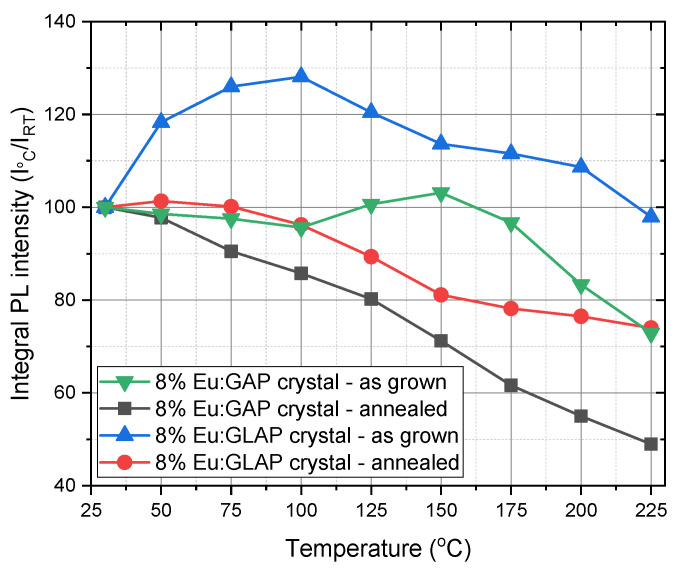
Temperature dependence of PL integral intensity of Eu:GAP and Eu:GLAP crystals with or without post Ar (+5% H_2_) annealing process, λ_ex_ = 275 nm.

**Figure 8 materials-16-00488-f008:**
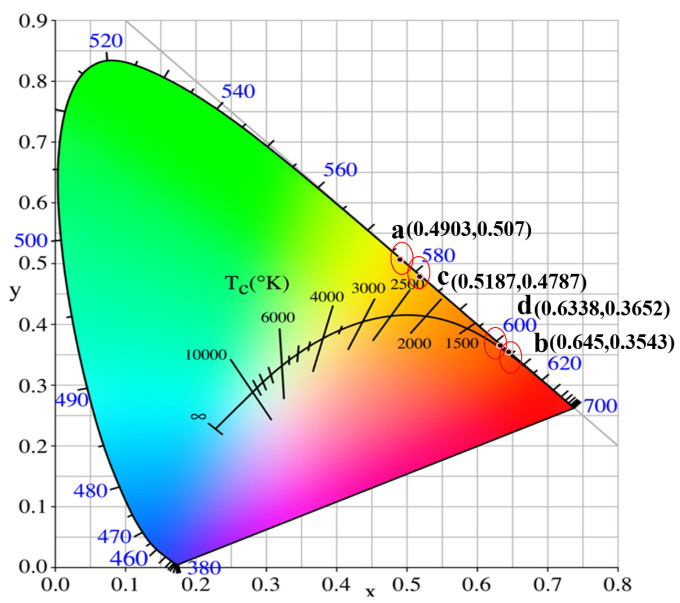
CIE chromaticity coordinates for the (**a**) as-grown Eu:GAP crystal; (**b**) annealed Eu:GAP crystal; (**c**) as-grown Eu:GLAP crystal; (**d**) annealed Eu:GLAP crystal.

**Figure 9 materials-16-00488-f009:**
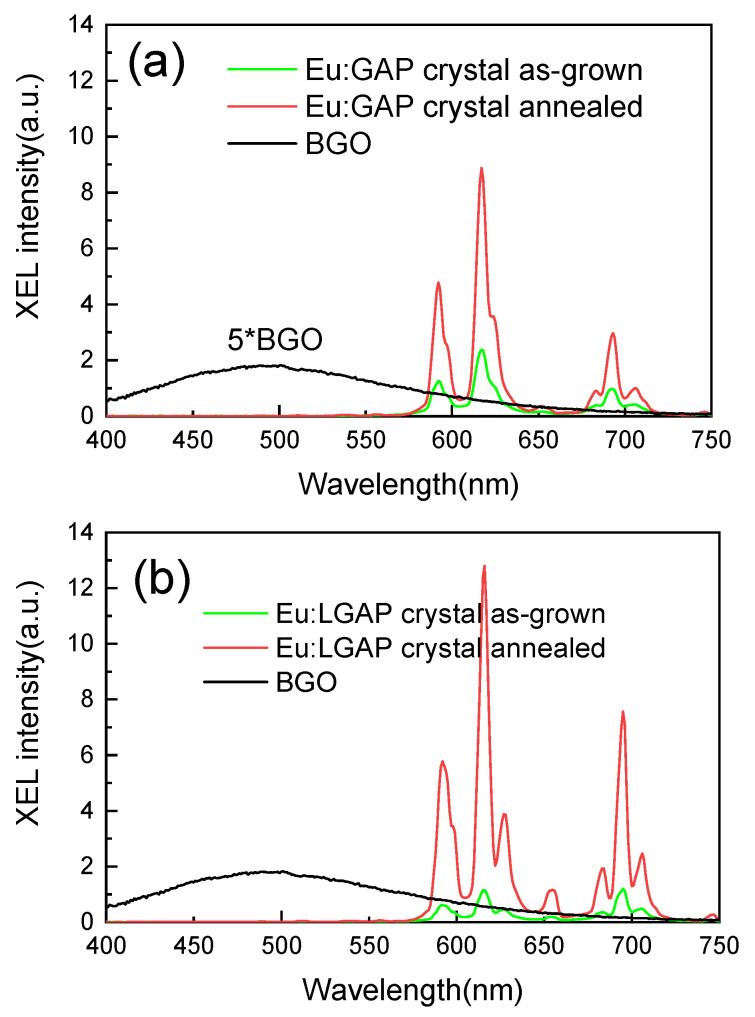
XEL spectra of (**a**) Eu:GAP and (**b**) Eu:GLAP crystals, with or without the 1000 °C for 4 h in Ar + 5% H_2_ annealing process. Five times the XEL intensity of a BGO single crystal was used for quantitative comparison.

**Table 1 materials-16-00488-t001:** Summarized color coordinates of Eu:GAP and Eu:GLAP crystals with or without the 1000 °C for 4 h in Ar + 5% H_2_ annealing process.

No.	Samples	CIE x	CIE y
a	Eu:GAP (as grown)	0.4903	0.5070
b	Eu:GAP (annealed)	0.6450	0.3543
c	Eu:GLAP (as grown)	0.5187	0.4787
d	Eu:GLAP (annealed)	0.6338	0.3652

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
