# Peer review of "Eu3+-Doped (Gd, La)AlO3 Perovskite Single Crystals: Growth and Red-Emitting Luminescence"

_materials, 2023, doi:10.3390/ma16020488_

Round 1

Reviewer 1 Report

The manuscript under revision presents results of luminescence characterisation of Eu3+ doped (Gd, La)AlO3 perovskite single crystals  grown by optical floating zone technique.  The sole merit of this work is in production of the crystals while the results are predictable and trivial.  Also, there is concern regarding the data quality.

For example, the absorption spectra shown in Fig.4 and their interpretation is questionable. There is obvious shift in the position of sharp absorption lines attributed to Eu3+ f-f transitions in the annealed and as-grown crystals. This is not possible; hence the measurements are unsound.

Origin of the strong peak at ca. 310 nm in excitation spectra (Fig.5a) needs to be explained.

The c comparison with BGO crystal is mentioned but not demonstrated in Fig. 7

The manuscript is difficult to read. The language of the manuscript requires significant improvement. Examples of apparent of apparent mistakes are listed below.  

Line 71: Its lower 71 defect concentration and fewer light scattering sources are more conductive ?? to achieve white light emitting with high light efficiency. – This is clearly wrong wording

the incensed ?? 556 and 655 nm emission of Eu:GAP and Eu:GLAP – what does this supposed to mean?

Awkward term “fluent atmosphere” used throughout the manuscript.

Summing up the manuscript does not merit publication in the journal

Reviewer 2 Report

The paper deals with an interesting topic because growing such good quality crystals is not easy. It deserves to be considered for publication in Materials, but after addressing all my comments. The language and the style could be improved (see the first comment). All the figures, tables, etc., are provided with satisfactory quality. In my opinion, the paper needs significant revision according to the comments below:

11.      Some sentences are unclear or have the wrong style and should be rewritten. For example: In lines 43-45, the sentence (In 2018, V. Gorbenko et al. 2 reported that….) , the same for 55-58 (However, the flexible composition of the GAP crystal….), 68-70 and more. I recommend checking the whole manuscript for such sentences.

22.      I would recommend explaining in the Introduction section in a more detailed way why the authors chose only 8 at. % of Eu, why this alloy is attractive etc.

33.       The accurate composition should be checked after growing. The composition along the crystal rod can vary; therefore, checking it along the growth axis is essential. The color of the grown crystals is changing, as can be seen in Figure 1a, so in my opinion, to properly analyze obtained experimental data, one should measure composition with some EDX/EDS method, for instance. It is known that many important properties depend on the composition (energy gap, optical and transport properties, lattice constant, etc.). Not only should axial homogeneity be checked, but I also recommend checking the radial homogeneity of the obtained plates. The composition could also be studied with XRD, so the question is, what was the place of the sample (position in crystal rod) used for XRD measurements? It would be interesting to see PXRD patterns for sample cut from the beginning and end of the crystal rod.

44.      The authors say about 8 atomic % of Eu3+ in GAP and GLAP crystals in the Introduction section. In lines 142-143, they say: “However, the narrow-spectrum absorption peak associated with the Eu3+ 4f - 4f transition observed in the 300 - 500 nm range is weaker due to the lower doping concentration of Eu3+”. 8 at. % is not doping; it is mixing already. Please clarify this issue.

55.       In the absorbance spectra, below the fundamental gap, the absorbance is still around 2 in the case of the GAP compound and 0.5 for GLAP. Well-polished samples should exhibit transparency of the order of 30-60%. What is the reason for reduced transparency?

66.      What do the authors mean by “La cooping” in line 159?

77.      Were all PLand PLE measurements conducted only at RT? It would be interesting to record the temperature evolution of such spectra. Absorption peaks visible in PLE spectra could be correlated with absorbance presented in Fig.4

According to the above comments, I recommend a major revision of the article.

Round 2

Reviewer 2 Report

After carefully studying, I have to say that the authors improved the paper significantly. They addressed all my comments; they only did not provide compositional analysis. But I agree with the authors this should be much more important in the case of several compositions. The strongest point of this work is the high-quality grown crystals. Therefore, in my opinion, this work deserves to be considered for publication. In summary, my recommendation is to accept the paper.